# Insulin Resistance Does Not Impair Mechanical Overload-Stimulated Glucose Uptake, but Does Alter the Metabolic Fate of Glucose in Mouse Muscle

**DOI:** 10.3390/ijms21134715

**Published:** 2020-07-01

**Authors:** Luke A. Weyrauch, Shawna L. McMillin, Carol A. Witczak

**Affiliations:** 1Department of Kinesiology, East Carolina University, Greenville, NC 27858, USA; weyrauchl16@students.ecu.edu (L.A.W.); mcmillins15@students.ecu.edu (S.L.M.); 2Department of Biochemistry & Molecular Biology, Brody School of Medicine, East Carolina University, Greenville, NC 27834, USA; 3Department of Physiology, Brody School of Medicine, East Carolina University, Greenville, NC 27834, USA; 4East Carolina Diabetes & Obesity Institute, East Carolina University, Greenville, NC 27834, USA; 5Department of Anatomy, Cell Biology & Physiology, Indiana University School of Medicine, Indianapolis, IN 46202, USA; 6IBRI Diabetes Center, Indiana Biosciences Research Institute, Indianapolis, IN 46202, USA

**Keywords:** exercise, glucose transporter, glycogen, glycolysis, hexosamine pathway, lactate, pentose phosphate pathway

## Abstract

Skeletal muscle glucose uptake and glucose metabolism are impaired in insulin resistance. Mechanical overload stimulates glucose uptake into insulin-resistant muscle; yet the mechanisms underlying this beneficial effect remain poorly understood. This study examined whether a differential partitioning of glucose metabolism is part of the mechanosensitive mechanism underlying overload-stimulated glucose uptake in insulin-resistant muscle. Mice were fed a high-fat diet to induce insulin resistance. Plantaris muscle overload was induced by unilateral synergist ablation. After 5 days, muscles were excised for the following measurements: (1) [^3^H]-2-deoxyglucose uptake; (2) glycogen; 3) [5-^3^H]-glucose flux through glycolysis; (4) lactate secretion; (5) metabolites; and (6) immunoblots. Overload increased glucose uptake ~80% in both insulin-sensitive and insulin-resistant muscles. Overload increased glycogen content ~20% and this was enhanced to ~40% in the insulin-resistant muscle. Overload did not alter glycolytic flux, but did increase muscle lactate secretion 40–50%. In both insulin-sensitive and insulin-resistant muscles, overload increased 6-phosphogluconate levels ~150% and decreased NADP:NADPH ~60%, indicating pentose phosphate pathway activation. Overload increased protein O-GlcNAcylation ~45% and this was enhanced to ~55% in the insulin-resistant muscle, indicating hexosamine pathway activation. In conclusion, insulin resistance does not impair mechanical overload-stimulated glucose uptake but does alter the metabolic fate of glucose in muscle.

## 1. Introduction

Exercise training is one of the most well-established therapeutic strategies for the treatment of progressive metabolic diseases such as type 2 diabetes [1,2,3,4,5,6,7,8]. Importantly, while previous work has demonstrated that resistance exercise training and chronic mechanical muscle loading (overload) stimulate glucose uptake into both insulin-sensitive and insulin-resistant skeletal muscle [9,10], the cellular or metabolic adaptations that underlie these beneficial effects are still largely unknown. 

Glucose uptake into skeletal muscle is regulated by the presence of cell surface glucose transporters as well as its metabolic fate, including: (1) storage as glycogen; (2) flux via glycolysis; (3) the pentose phosphate pathway; and (4) the hexosamine pathway. Importantly, studies have linked excessive muscle glycogen accumulation and increased glucose flux through the hexosamine pathway with the development of muscle insulin resistance [11,12,13,14,15], highlighting the importance of glucose metabolism in regulating future increases in muscle glucose uptake. In addition, changes in glucose metabolism such as increased glycolytic flux and activation of the pentose phosphate pathway have been linked with enhanced cellular growth in a variety of cell types [16,17,18,19]. Despite the beneficial effects of resistance training/overload on systemic glucose homeostasis and muscle glucose uptake, little is known regarding its effects on muscle glucose metabolism. Previous work has shown that resistance training/overload results in small but inconsistent (0–45%) increases in muscle glycogen levels [6,9,10,20,21,22,23], while chronic muscle loading has been shown to increase glycolytic flux (30–60%) [9]. Whether overload directs glucose flux into other metabolic pathways, and whether this partitioning of glucose is affected by insulin resistance, has not yet been examined. 

The primary purpose of this study was to determine if the ability of overload to stimulate glucose uptake in insulin-resistant skeletal muscle is associated with a differential partitioning of glucose into metabolic pathways. We hypothesized that: (1) glycogen content and glycogen synthesis rates would be increased by overload, and not affected by insulin resistance; (2) glycolytic flux and pentose phosphate pathway activity would be increased by overload and not affected by insulin resistance; and (3) hexosamine pathway activity would be increased in insulin-resistant muscle, and decreased by overload. The main findings from this study demonstrate that mechanical overload stimulates an increase in muscle glycogen synthesis, lactate secretion, the pentose phosphate pathway and the hexosamine pathway. Insulin resistance impaired overload-stimulated muscle glycogen synthesis; did not alter the overload-stimulated increase in lactate secretion, or activation of the pentose phosphate pathway; but enhanced the overload-stimulated activation of the hexosamine pathway. Collectively, these findings highlight the importance of glucose partitioning as part of the mechanosensitive mechanism underlying overload-stimulated muscle glucose uptake.

## 2. Results

### 2.1. High-Fat Diet-Induced Mouse Model of Insulin Resistance

To determine whether insulin resistance impairs mechanical overload-stimulated skeletal muscle glucose uptake and/or glucose metabolism, a rodent model of high-fat diet-induced insulin resistance was utilized. Male C57BL/6J mice were fed either a low-fat diet (LFD) or high-fat diet (HFD) ad libitum for 12 weeks. Mice fed the HFD exhibited higher body weights (~60%), fasted blood glucose levels (~70%), fasted serum insulin levels (~450%) and HOMA-IR values (~900%) compared to LFD controls, demonstrating the development of obesity and whole body insulin resistance (Figure 1A–D). To assess the development of skeletal muscle insulin resistance, isolated plantaris muscles were incubated in buffer containing [^3^H]-2-deoxyglucose in the absence or presence of a submaximal dose of insulin. Insulin increased glucose uptake ~100% in muscles from the LFD mice, but only ~60% in muscles from the HFD mice (Figure 1E), demonstrating that the HFD induced plantaris muscle insulin resistance. 

### 2.2. Insulin Resistance Does Not Impair Overload-Stimulated Muscle Glucose Uptake

To determine whether insulin resistance impairs overload-stimulated muscle glucose uptake, mice underwent unilateral synergist ablation surgery to induce plantaris muscle overload. After 5 days, muscles were removed and immediately weighed. Overload increased plantaris muscle weight ~40% in LFD mice, and this overload-stimulated increase in muscle weight was ~2% greater in HFD mice (Figure 2A). To determine whether insulin resistance impairs overload-stimulated glucose uptake, muscles were incubated in buffer containing [^3^H]-2-deoxyglucose in the absence of insulin. Overload increased glucose uptake ~80% in muscles from both the LFD and HFD mice (Figure 2B), demonstrating that insulin resistance does not impair overload-induced muscle glucose uptake.

### 2.3. Insulin Resistance Does Not Impair Overload-Stimulated Glucose Transporter Expression

Glucose uptake into muscle requires both transport across the cell surface by glucose transporter (GLUT) proteins as well as phosphorylation by the enzyme hexokinase. Strikingly, recent work demonstrated that the major GLUT isoform expressed in skeletal muscle, GLUT4, is not necessary for overload-stimulated muscle glucose uptake [24]. Instead, studies have shown that overload stimulates an increase in the mRNA and protein levels of GLUT1, GLUT3, GLUT6, and GLUT10 in insulin-sensitive mouse muscle [24,25]. To determine whether insulin resistance alters the effects of overload on GLUT isoform or hexokinase II protein levels, immunoblots were performed (Figure 3A–F). While overload did not increase GLUT1 protein levels in muscles from LFD mice, it increased GLUT1 levels ~70% in muscles from HFD mice (Figure 3A). Overload increased GLUT3 levels ~65% in muscles from LFD mice, and increased GLUT3 levels more in muscles from the HFD mice (~100%; Figure 3B). Overload increased GLUT6 levels ~350% and GLUT10 levels ~35% in muscles from both LFD and HFD mice (Figure 3D,E). Neither GLUT4 nor hexokinase II protein levels were altered by overload in muscles from either the LFD or HFD mice (Figure 3C,F). Collectively these findings suggest that increases in GLUT1, GLUT4 or hexokinase II protein levels are not part of the mechanism underlying overload-stimulated muscle glucose uptake. 

### 2.4. Insulin Resistance Enhances Overload-Stimulated Accumulation of Muscle Glycogen

Skeletal muscle glycogen content is inversely related to glucose uptake [26]. To determine whether lower glycogen levels are part of the mechanism underlying overload-stimulated muscle glucose uptake, muscle glycogen levels were assessed using a hexokinase-based reagent. Overload increased glycogen content ~25% in muscles from LFD mice and this increase was greater (~40%) in muscles from HFD mice (Figure 4A), demonstrating that lower glycogen content is not part of the mechanism underlying overload-stimulated glucose uptake. To determine whether the higher glycogen content observed in the overload-stimulated muscles was due to increased glycogen synthesis, muscles were incubated in [5-3H]-glucose for 1 h, and the incorporation of the radiolabel into glycogen (i.e., [5-3H]-glycogen) assessed. In response to overload, glycogen synthesis increased ~25% in muscles from LFD mice, but only trended towards a significant increase (~12%; *p* = 0.058) in muscles from HFD mice (Figure 4B). Thus, the higher glycogen content seen in the overload-stimulated muscles from the HFD mice is not due to an increase in glycogen synthesis rates.

### 2.5. Overload Does Not Alter Glycolytic Flux in Insulin Sensitive or Insulin Resistant Muscle

Glucose is utilized in several major metabolic pathways, and the partitioning of glucose into these different pathways can have critical consequences for future increases in muscle glucose uptake. Glycolysis is a key glucose utilizing pathway in skeletal muscle as it plays a critical role in generating ATP under both oxygen abundant and hypoxic conditions. To determine whether increased glucose flux through glycolysis is part of the mechanism underlying overload-stimulated muscle glucose uptake, muscles were incubated in [5-^3^H]-glucose and the generation of [^3^H]-H_2_O from the enolase reaction utilized to calculate muscle glycolytic flux rate. As shown in Figure 5A, overload did not increase glycolytic flux in muscles from either LFD or HFD mice. To further assess glucose utilization via glycolysis downstream of the enolase reaction, buffer lactate levels were measured as an indicator of muscle lactate secretion. As shown in Figure 5B, overload increased muscle lactate secretion 40–50% in muscles from both LFD and HFD mice. 

### 2.6. Overload-Stimulated Activation of the Pentose Phosphate Pathway Is Not Impaired by Insulin Resistance

The pentose phosphate pathway is a glucose utilizing metabolic pathway that generates molecules critical for muscle growth, including NADPH for reductive biosynthesis reactions and key metabolic precursors for nucleotide and amino acid synthesis. To assess whether activation of the pentose phosphate pathway could be part of the mechanism underlying overload-stimulated muscle glucose uptake, the protein levels of the initiating enzyme of this pathway, glucose-6-phosphate dehydrogenase (G6PD), were assessed by immunoblot analysis. As shown in Figure 6A, overload increased G6PD levels ~140% in muscles from LFD mice, and this effect was attenuated in muscles from HFD mice (only a ~80% increase). To further investigate the effects of insulin resistance and overload on the pentose phosphate pathway, several key pentose phosphate pathway metabolites were assessed by UPLC-MRM/MS. In muscles from both LFD and HFD mice, overload increased 6-phosphogluconate levels (140–160%; Figure 6B). In addition, overload decreased NADP levels ~40% and increased NADPH levels ~65% in muscles from LFD mice, and decreased NADP levels ~55% and increased NADPH levels ~20% in muscles from the HFD mice (Figure 6C,D). When evaluated together, overload decreased the ratio of NADP:NADPH ~60% in muscles from both LFD and HFD mice (Figure 6E). Collectively, these findings suggest that overload activates the pentose phosphate pathway in skeletal muscle and that this activation is not impaired by insulin resistance.

### 2.7. Overload-Stimulated Activation of the Hexosamine Pathway Is Enhanced by Insulin Resistance

The hexosamine pathway is an additional glucose utilizing metabolic pathway that generates UDP-N-acetylglucosamine and other nucleotide hexosamines for the O-linked GlcNAcylation of proteins. To assess whether activation of the hexosamine pathway could be part of the mechanism underlying overload-stimulated muscle glucose uptake, immunoblots were performed to assess the protein levels of the initiating enzyme of this pathway, glutamine fructose-6-phosphate transaminase 1 (GFPT1). Overload increased GFPT1 levels ~50% in muscles from both LFD and HFD mice (Figure 7A). To further investigate the effects of insulin resistance and overload on the hexosamine pathway in skeletal muscle, immunoblots were performed to assess total muscle protein O-GlcNAcylation. As shown in Figure 7B, overload increased protein O-GlcNAcylation ~45% in muscles from LFD mice, and this effect was greater in muscles from the HFD mice (increased ~55%). Thus, collectively these findings suggest that overload activates the hexosamine pathway in skeletal muscle and that this effect is enhanced by insulin resistance.

## 3. Discussion

The major findings from this study demonstrate that HFD-induced insulin resistance impairs overload-stimulated muscle glycogen synthesis; does not alter overload-stimulated increases in lactate secretion or activation of the pentose phosphate pathway; but does enhance the overload-stimulated activation of the hexosamine pathway. Collectively, these findings highlight the importance of glucose metabolism as part of the mechanosensitive mechanism underlying overload-stimulated muscle glucose uptake.

The glucose transporter that mediates overload-stimulated muscle glucose uptake is presently unknown. However, findings from this study demonstrate that overload increases GLUT1, GLUT3, GLUT6 and GLUT10 protein levels in insulin resistant muscle (Figure 3), suggesting that one or more of these GLUT isoforms may be responsible for this effect. Surprisingly, in this study, overload failed to significantly increase GLUT1 levels in muscles from the LFD mice (Figure 3A). The reason underlying this difference in finding is presently not clear, but is likely not due to the sex of mice, as we have previously observed a significant increase in GLUT1 protein levels following overload in muscles from both male and female mice [24]. The abundance of GLUT3, GLUT6 or GLUT10 protein in muscle relative to the abundance of other GLUT isoforms has not yet been determined. However, previous work has demonstrated that GLUT3 is localized to the cell surface in myotubes and has a high affinity for 2-deoxyglucose (K_m_ = 1.8 mM) and d-glucose (K_m_ = 1.4 mM) [27,28,29], raising the possibility that the 65–100% increase in GLUT3 levels induced by overload could lead to a substantial increase in muscle glucose uptake. The subcellular localization of GLUT6 and GLUT10 in skeletal muscle has not yet been investigated. However previous work has demonstrated that GLUT6 can be localized to both the cell surface and an intracellular compartment (e.g., lysosome) depending on the cell type [30,31], while to date GLUT10 has been detected primarily in mitochondria in smooth muscle cells [32]. Future studies assessing the localization of these GLUTs in skeletal muscle in response to overload as well as studies utilizing novel muscle-specific GLUT isoform knockout mice are needed to definitively determine the GLUT responsible for overload-stimulated muscle glucose uptake.

In this study, overload stimulated glycogen accumulation 25–45% in plantaris muscles from the LFD and HFD fed mice (Figure 4A). This increase in muscle glycogen content was consistent with several studies from both humans and rats that demonstrated a 15–45% increase in muscle glycogen content following 6–16 weeks of resistance exercise training [10,20,23,33]; whereas they were in contrast with other 6-week resistance training studies in humans that demonstrated no effect of muscle glycogen levels [21,22]. These findings are also in contrast with previous work that demonstrated that 4 days of overload was not sufficient to increase glycogen content in soleus muscles from healthy or chow diet overeating-induced insulin resistant mice [9]. The reason underlying this discrepancy in findings is presently unknown but could be due to differences in mouse strain (i.e., C57BL/6J vs. Swiss albino) and/or muscle examined (i.e., plantaris vs. soleus).

Muscle protein synthesis is a high ATP demanding metabolic process. Thus, we hypothesized that mechanical overload might increase glucose utilization through glycolysis to fuel the increased energetic demand of protein synthesis and muscle hypertrophic growth. Intriguingly, while overload did not increase muscle glycolytic flux through the enolase reaction (Figure 5A); it did increase lactate secretion in muscles from both the LFD and HFD mice (Figure 5B). Taken together these results could possibly suggest that overload decreases muscle glucose oxidation. Unfortunately, in this study measures of glucose oxidation were not possible due to the inability to capture labelled CO_2_ gas with the current muscle incubation apparatus. Future studies will need to be performed to address the contribution of substrate oxidation to overload-stimulated changes in glucose metabolism in both insulin-sensitive and insulin-resistant muscles.

In both insulin-sensitive and insulin-resistant muscle, overload increased G6PD protein levels ~140%, 6-phosphogluconate levels 140–160% and NADPH levels 20–65% (Figure 6), suggesting that overload stimulates an increase in glucose flux through the pentose phosphate pathway in skeletal muscle. The results from this study are consistent with work performed in rat skeletal muscle that demonstrated a 100–350% increase in G6PD activity 2 days after a single bout of downhill treadmill running [34], as well as an ~20% increase in ribose-5-phosphate levels immediately after 10 min of high intensity tetanic contractions [35]. Thus, collectively these findings support the idea that the pentose phosphate pathway is an exercise- and overload-responsive glucometabolic pathway in muscle. Future work is needed to directly determine the amount of glucose that is redirected into this pathway in muscle following overload and resistance exercise training.

While activation of the hexosamine pathway has been linked to the development of insulin resistance [15,36], inhibition of this pathway has been linked to an increase in insulin sensitivity [36]. Thus, we hypothesized that the HFD would increase GFPT1 and total protein O-GlcNAcylation levels in muscle, and that these effects would be reversed by overload. In contrast, the results of this study demonstrated that while muscles from the HFD mice did not have increased GFPT1 nor increased total protein O-GlcNAcylation levels (Figure 7), both GFPT1 and O-GlcNAcylation were increased by overload (Figure 7). Thus, additional studies should be performed to determine which proteins are O-GlcNAcylated in muscle in response to overload to fully assess the role of this pathway to chronic loading-stimulated muscle adaptations.

The glucose uptake and glucose metabolism measurements presented in this study were generated using an ex vivo skeletal muscle model system. This model system was chosen because it provides the ability to control substrate availability (i.e., extracellular glucose concentrations) while eliminating the possible contribution of systemic factors (e.g., changes in blood flow, neuronal activity or circulating hormones such as insulin). Thus, by design this model does not fully recapitulate the multitude of factors that could influence loading-stimulated muscle glucose uptake or glucose metabolism in the in vivo environment. Therefore, the findings from this study should be viewed as a crucial starting point for future studies that would assess loading-stimulated changes in muscle metabolism in living animal or human subjects that were both insulin sensitive and insulin resistant. 

## 4. Materials and Methods

### 4.1. Animals

Experiments were performed in accordance with the East Carolina University Institutional Animal Care and Use Committee (Animal Use Protocol #s: P066a and P066b) and the National Institutes of Health Guidelines for the Care and Use of Laboratory Animals. Male C57BL/6J mice (~6 weeks old) were obtained from The Jackson Laboratory (Bar Harbor, ME, USA) and fed either a standard chow, 14% kcal low-fat diet (LFD; Prolab®RMH3000, cat#5P75*, PMI Nutritional International, St. Louis, MO, USA) or a 60% kcal high-fat diet (HFD; cat#D12492, Research Diets, New Brunswick, NJ, USA) for 12 weeks. Only male mice were utilized in this study because previous work has indicated that male mice exhibit greater obesity, glucose intolerance and insulin resistance in response to high-fat feeding compared to female mice (data not shown) [37,38,39,40]. Male CD-1 mice (~6 weeks old) were obtained from Charles River Laboratories and provided standard chow. Food and water were available ad libitum. Mice were housed in cages at 21–22 °C with a 12 h light/dark cycle. 

### 4.2. Systemic Insulin Resistance Assessment

After 12 weeks of diet intervention, mice were fasted overnight (12–14 h), weighed, and blood samples collected from the tail vein to assess glucose levels using a glucometer (OneTouch Ultra2, LifeScan Inc., Milpitas, CA, USA), and insulin levels by enzyme-linked immunosorbent assay (cat#EZRMI-13K, EMD Millipore, Burlington, MA, USA). The homeostatic model assessment of insulin resistance (HOMA-IR) was calculated by multiplying fasted blood glucose (mM) by fasted serum insulin (mU/L) and dividing by 22.5. 

Mice that did not clearly segregate into an LFD, insulin-sensitive or HFD, insulin-resistant group were excluded from further experiments (excluded: *N* = 14/60 LFD mice; *N* = 13/60 HFD mice). All remaining mice were randomly assigned to one of the following experimental groups: (1) insulin-stimulated muscle glucose uptake; (2) overload-stimulated muscle glucose uptake and total glycogen; (3) intracellular protein expression levels; (4) glycolytic flux, lactate secretion and glycogen synthesis; or (5) pentose phosphate pathway metabolites. 

### 4.3. Unilateral Synergist Ablation Surgery

Plantaris muscle mechanical overload was induced via unilateral ablation of synergist muscles using methods previously described by our lab [24,41]. Mice were anaesthetized with isoflurane (2–3%) and the distal two-thirds of the gastrocnemius and soleus surgically ablated. A sham surgery was performed on the contralateral leg. After 5 days, mice were fasted overnight, anesthetized with pentobarbital sodium (90–100 mg/kg body weight) for 40 min and euthanized by cervical dislocation. Muscles were excised and weighed to ±0.1 g using an analytical balance (Mettler-Toledo XS105, Columbus, OH, USA), prior to additional experimental perturbations as described below. 

### 4.4. Muscle [^3^H]-2-Deoxyglucose Uptake 

Insulin- and overload-stimulated [^3^H]-2-deoxyglucose uptake was assessed in isolated muscles using methods previously described by our lab [24,42]. For insulin-stimulated glucose uptake, muscles were incubated in continuously gassed (95% O_2_, 5% CO_2_), 37 °C Krebs-Ringer-Bicarbonate (KRB) buffer containing: 117 mM NaCl, 4.7 mM KCl, 2.5 mM CaCl_2_, 1.2 mM KH_2_PO_4_, 1.2 mM MgSO_4_, and 24.6 mM NaHCO_3_ supplemented with 2 mM pyruvate for 70 min, and incubated in the presence or absence of submaximal insulin (600 µU/mL = 4167 pM; cat#11376497001, Sigma-Aldrich, St. Louis, MO, USA) for 20 min. For overload-stimulated glucose uptake, muscles were incubated in continuously gassed, 37 °C KRB buffer+ 2 mM pyruvate for 90 min. To assess glucose uptake, muscles were incubated in continuously gassed, 30 °C KRB buffer containing 1.5 µCi/mL [^3^H]-2-deoxy-d-glucose (cat#NET54900, Perkin Elmer, Waltham, MA, USA), 1 mM 2-deoxy-d-glucose, 0.45 µCi/mL [^14^C]-mannitol (Perkin Elmer, cat#NEC314), and 7 mM mannitol for 10 min. Muscles were frozen in liquid nitrogen. Frozen muscles were weighed, solubilized in 1.0 N NaOH at 80 °C for 15 min, and the solution neutralized with 1.0 N HCl. For overload-stimulated muscle samples, an aliquot was removed to assess total muscle glycogen as described below. The remaining fraction was centrifuged at 10,000× *g* for 1 min. Aliquots were removed for scintillation counting of the [^3^H] and [^14^C] labels, and [^3^H]-2-deoxyglucose uptake calculated.

### 4.5. Muscle Glycogen Content

Glycogen content was determined from aliquots of muscle lysate taken from ex vivo overload-stimulated glucose uptake experiments as described above. Aliquots were incubated in 3.33 M HCl at 95 °C for 120 min and neutralized by the addition of 2 M NaOH and 1.5 M Tris (pH 7.8). A hexokinase-based reagent (cat#2820-3, Eagle Diagnostics, Cedar Hill, TX, USA) was added, and glycogen content measured spectrophotometrically at 340 nm using a SpectraMax M4 multimode plate reader (Molecular Devices, San Jose, CA, USA). 

### 4.6. Muscle Glycolytic Flux and Lactate Secretion

Glycolytic flux rates were assessed in isolated muscles by measuring the rate of [5-^3^H]-glucose conversion to [^3^H]-H_2_O using methods adapted from those previously described [43]. Plantaris muscles were incubated in continuously gassed, 37 °C KRB + 2 mM pyruvate buffer for 90 min, and KRB buffer containing 2.75 µCi/mL d-[5-^3^H]-glucose (cat# NET531, Perkin Elmer, Waltham, MA, USA) and 5.5 mM d-glucose for 60 min. Muscles were removed and frozen in liquid nitrogen to assess glycogen synthesis as described below. Aliquots of buffer were collected to assess glycolytic flux and muscle lactate secretion. For glycolytic flux, [5-^3^H]-glucose was separated from [^3^H]-H_2_O using Dowex-1 borate mini-columns. Dowex-1 borate was generated by washing Dowex 1 × 8 chloride (cat#44340, Millipore Sigma, St. Louis, MO, USA) with sodium tetraborate. [^3^H]-H_2_O levels were assessed by liquid scintillation counting, and glycolytic flux rates calculated and normalized to pre-incubation muscle weights. For muscle lactate secretion, buffer lactate levels were assessed using a lactate colorimetric/fluorometric assay kit (cat# K607-100, Biovision Inc., Milpitas, CA, USA).

### 4.7. Muscle Glycogen Synthesis

Muscle glycogen synthesis rates were assessed by measuring the rate of [5-^3^H]-glucose incorporation into glycogen using methods previously described [44]. Frozen muscles from the glycolytic flux experiment were solubilized in 30% KOH at 95 °C for 20 min. Glycogen was precipitated by the addition of 50 µL of 5% glycogen (Sigma-Aldrich, cat#G8876), 100% EtOH (final concentration = 70%), and incubating at −20 °C overnight. Samples were centrifuged at 8900× *g* for 20 min to pellet glycogen. Pellets were washed with 66% EtOH and dissolved in deionized water. Samples were spotted onto P81 grade ion exchange chromatography paper (cat#3698-325, GE Whatman, Marlborough, MA, USA), washed with 66% EtOH, and air-dried. [5-^3^H]-glycogen was measured via liquid scintillation counting, and glycogen synthesis rates calculated and normalized to muscle weight. 

### 4.8. Pentose Phosphate Pathway Metabolites

Whole frozen plantaris muscles were sent to Creative Proteomics (Shirley, NY, USA) for the quantification of NADP, NADPH, and 6-phosphogluconate levels. Muscles were weighed and then homogenized in cold 80% aqueous MeOH using a MM 400 mill mixer 2 × 1 min. Samples were centrifuged at 21,130× *g* at 5 °C, and 400 µL of clear supernatant removed and mixed with 100 µL of a 13C10-GTP (internal standard) solution and 250 µL chloroform. The mixture was vortexed for 1 min and then centrifuged at 21,130× *g* at 5 °C for 10 min. The upper aqueous phase was removed and dried under a nitrogen gas flow. The residue was reconstituted in 200 µL of 50% acetonitrile. A 10-µL aliquot of each sample was injected into a HILIC (hydrophilic interaction liquid chromatography) column (2.1 × 150 mm, 1.7 µm) for gradient elution with ammonium buffer (mobile phase A) and acetonitrile (mobile phase B). The efficient gradient was from 75% B to 15% B in 12 min, with a flow rate of 300 µL/min and a column temperature of 30 °C. Metabolites were detected by UPLC-MRM/MS with (-) ion detection using a Dionex 3400 UHPLC system coupled to a 4000 QTRAP mass spectrometer. 

### 4.9. Immunoblot Analyses

Immunoblot analyses were performed using standard methods as described by our lab [24,41,42]. Frozen muscles were homogenized in buffer containing: 20 mM Tris-HCl pH 7.5, 5 mM EDTA, 10 mM Na_4_P_2_O_7_, 100 mM NaF, 2 mM NaVO_4_, 0.0015 mM aprotinin, 0.01 mM leupeptin, 3 mM benzamidine, 1 mM phenylmethylsulfonylfluoride, and 1% IGEPAL^®^ CA-630. Samples were rotated end-over-end at 4 °C for 60 min and centrifuged at 13,000× *g* for 30 min. Lysate protein concentrations were determined via the Bradford method using Bio-Rad protein assay dye (cat#5000006, Bio-Rad Laboratories, Hercules, CA, USA). Lysates (40 µg) were subjected to SDS-PAGE, and proteins transferred onto 0.2 µm nitrocellulose membranes. Ponceau S solution (cat#P7170, Sigma-Aldrich, St. Louis, MO, USA) was utilized to assess equal protein loading and transfer. Immunoblotting conditions were as described in Table 1. Densitometric analysis of immunoblots was performed using a Bio-Rad Chemidoc XRS+ imaging system and Bio-Rad Image Lab™ software v 6.0.1 (Bio-Rad Laboratories, Hercules, CA, USA).

Glucose transporter (GLUT) antibodies were previously validated by our group using mouse tibialis anterior muscle samples overexpressing individual mouse GLUT isoforms [24]. Glucose-6-phosphate dehydrogenase (G6PD) and glutamine fructose-6-phosphate transaminase 1 (GFPT1) antibodies were validated using G6PDx or GFPT1 mouse muscle overexpression samples that were generated by transfecting the tibialis anterior muscles of male CD-1 mice (8 weeks old) with empty vector, untagged mouse G6PDx (cat#MC205271, OriGene Technologies, Rockville, MD, USA), or untagged mouse GFPT1 (cat#MC201036, OriGene Technologies, Rockville, MD, USA) vectors followed by standard in vivo muscle gene transfer/ electroporation procedures previously described by our lab [41,42]. After 1 week, muscles were collected and processed for immunoblot analyses as described above.

### 4.10. Statistical Analyses.

Data were graphed using GraphPad Prism v8 software (GraphPad Software Inc., San Diego, CA, USA) and are presented as individual data points with the mean ± standard deviation. Statistical significance was assessed using SigmaPlot v12 software (Systat Software Inc., San Jose, CA, USA) and defined as *p* < 0.05 determined by Student’s *t*-tests or Two-way Analysis of Variance and Tukey’s post hoc analysis. The number of mice or muscles utilized to determine statistical significance is indicated in the text or figure legends.

## 5. Conclusions

In conclusion, the results from this study demonstrate that while insulin resistance does not impair mechanical overload-stimulated increases in skeletal muscle glucose uptake, increases in lactate secretion, activation of the pentose phosphate pathway or activation of the hexosamine pathway; it does impair overload-stimulated increases in muscle glycogen synthesis. Future studies are currently underway to determine the GLUT isoform responsible for overload-stimulated increases in glucose uptake, as well as to quantify the amount of glucose that is partitioned into each glucometabolic pathway in response to overload. 

## Figures and Tables

**Figure 1 ijms-21-04715-f001:**
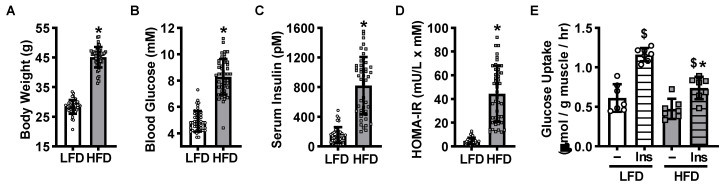
High-fat diet-induced mouse model of obesity and plantaris muscle insulin resistance. Male C57BL/6J mice were fed a low-fat diet (LFD) or high-fat diet (HFD) for 12 weeks. Following an overnight fast, mice were weighed (**A**). Blood was collected from the tail to assess glucose (**B**) and insulin levels (**C**). The homeostatic model assessment of insulin resistance (HOMA-IR) was calculated (**D**). Plantaris muscles were excised and [^3^H]-2-deoxy-d-glucose uptake assessed in the absence (**–**) or presence of submaximal insulin (Ins; 600 µU/mL = 4167 pM) (**E**). Data are mean ± standard deviation. Statistical analysis was performed using unpaired t-tests (**A**–**D**; *N* = 40–47 mice) or a 2-way ANOVA with Tukey’s post-hoc analysis (**E**; *N* = 7 muscles/group). *p* < 0.05 * vs. LFD; $ vs. **–**.

**Figure 2 ijms-21-04715-f002:**
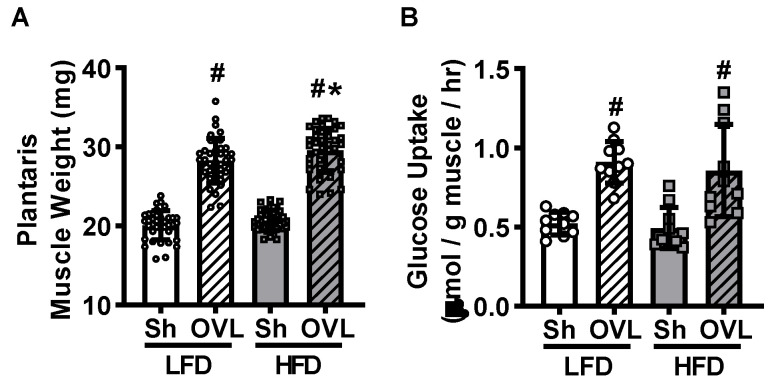
Insulin resistance does not impair overload-stimulated increases in plantaris muscle mass or glucose uptake. After 12 weeks of high-fat diet (HFD) feeding, mice underwent unilateral synergist ablation surgery to induce overload (OVL) of the plantaris muscle. Five days later isolated plantaris muscles were weighed (**A**), and [^3^H]-2-deoxy-d-glucose uptake assessed in the absence of insulin (**B**). Data are mean ± standard deviation. Statistical analysis was performed using a 2-way ANOVA with Tukey’s post-hoc analysis. *N* = 38–40 muscles/group (**A**), or 10 muscles/group (**B**). *p* < 0.05 # vs. sham (Sh); * vs. low-fat diet (LFD).

**Figure 3 ijms-21-04715-f003:**
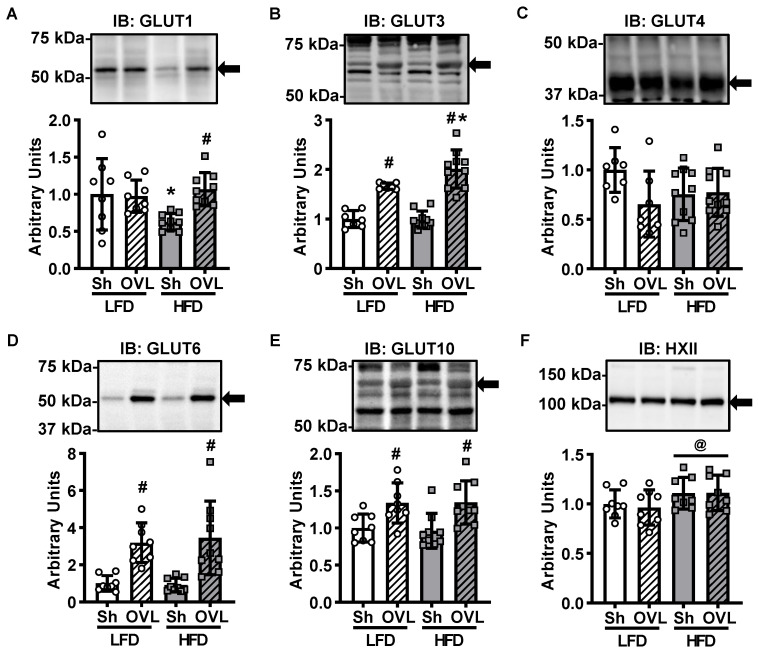
Insulin resistance does not impair overload-stimulated increases in glucose transporter protein levels. After 12 weeks of high-fat diet (HFD) feeding, mice underwent unilateral synergist ablation surgery to induce overload (OVL) of the plantaris muscle. Five days later, muscles were processed for immunoblot (IB) analysis of glucose transporter 1 (**A:** GLUT1); (**B**) GLUT3; (**C**) GLUT4; (**D**) GLUT6; (**E**) GLUT10; and (**F**) hexokinase II (HXII). Data are mean ± standard deviation. Statistical analysis was performed using a 2-way ANOVA with Tukey’s post-hoc analysis. *N* = 7–10 muscles/group. *p* < 0.05 # vs. Sham (Sh); * vs. low-fat diet (LFD); @ = main effect.

**Figure 4 ijms-21-04715-f004:**
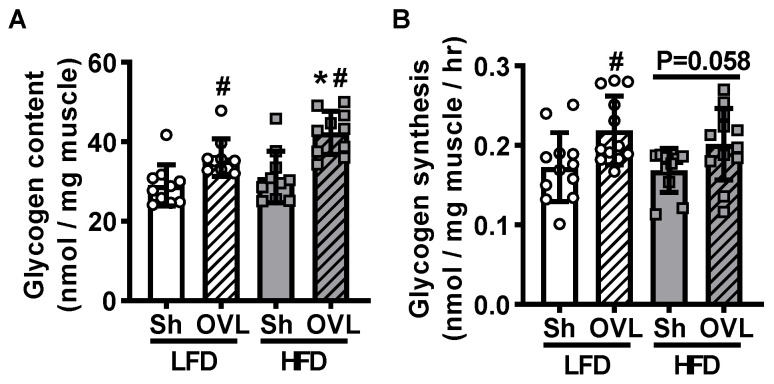
Insulin resistance enhances overload-stimulated muscle glycogen accumulation. After 12 weeks of high-fat diet (HFD) feeding, mice underwent unilateral synergist ablation surgery to induce overload (OVL) of the plantaris muscle for 5 days. (**A**) Total muscle glycogen content was assessed using a hexokinase-based assay; or (**B)** muscles were incubated in [5-^3^H]-glucose to determine glycogen synthesis. Data are mean ± standard deviation. Statistical analysis was performed using a 2-way ANOVA with Tukey’s post-hoc analysis. *N* = 10–12 muscles/group. *p* < 0.05 # vs. Sham (Sh); * vs. low-fat diet (LFD).

**Figure 5 ijms-21-04715-f005:**
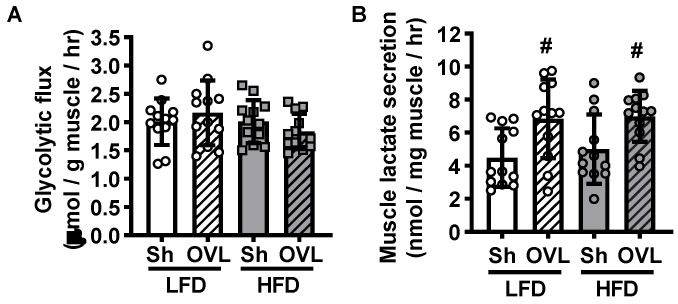
Insulin resistance does not impair muscle glycolytic flux, or overload-stimulated muscle lactate secretion. After 12 weeks of high-fat diet (HFD) feeding, mice underwent unilateral synergist ablation surgery to induce overload (OVL) of the plantaris muscle for 5 days. (**A**) Muscles were incubated in [5-^3^H]-glucose to determine glucose flux through glycolysis. (**B**) Buffer lactate levels were measured at the end of the glycolytic flux assay as an indicator of muscle lactate secretion. Data are mean ± standard deviation. Statistical analysis was performed using a 2-way ANOVA with Tukey’s post-hoc analysis. *N* = 12 muscles/group. *p* < 0.05 # vs. Sham (Sh).

**Figure 6 ijms-21-04715-f006:**
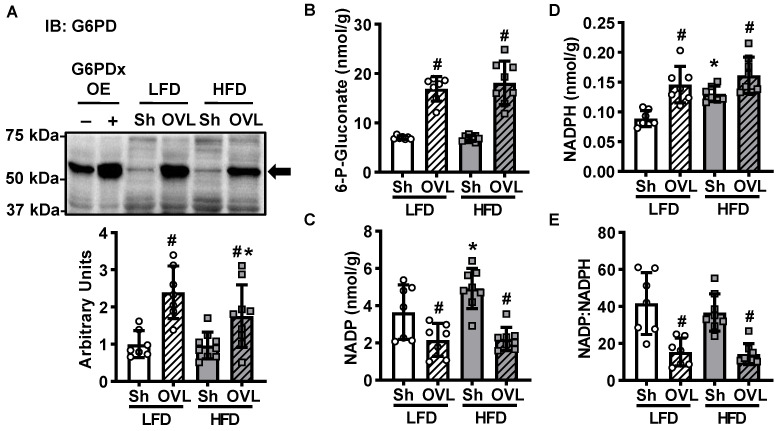
Insulin resistance does not impair overload-stimulated activation of the pentose phosphate pathway. After 12 weeks of high-fat diet (HFD) feeding, mice underwent unilateral synergist ablation surgery to induce overload (OVL) of the plantaris muscle for 5 days. (**A**) Muscles were excised and the protein content of glucose-6-phosphate dehydrogenase (G6PD) determined by immunoblot (IB) analysis. The G6PD antibody was validated using mouse tibialis anterior muscle samples overexpressing G6PDx (OE) or empty vector-transfected controls (-). (**B**–**D**) Ultra performance liquid chromatography/ multiple reaction monitoring-mass spectrometry (UPLC/MRM-MS) was used to measure 6-phosphogluconate, NADP and NADPH. (**E**) The ratio of NADP:NADPH was calculated. Data are mean ± standard deviation. Statistical analysis was performed using a 2-way ANOVA with Tukey’s post-hoc analysis. *N* = 7–9 muscles/group. *p* < 0.05 # vs. Sham (Sh); * vs. low-fat diet (LFD).

**Figure 7 ijms-21-04715-f007:**
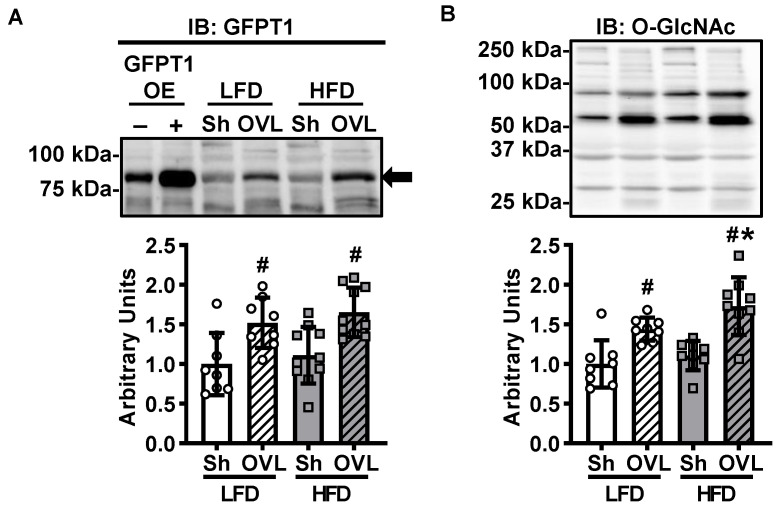
Insulin resistance enhances overload-stimulated activation of the hexosamine biosynthetic pathway. After 12 weeks of high-fat diet (HFD) feeding, mice underwent unilateral synergist ablation surgery to induce overload (OVL) of the plantaris muscle for 5 days. (**A**) Muscles were excised and the protein content of glutamine fructose-6-phosphate transaminase 1 (GFPT1) determined by immunoblot (IB) analysis. The GFPT1 antibody was validated using mouse tibialis anterior muscle samples overexpressing GFPT1 (OE) or empty vector-transfected controls (-). (**B**) Immunoblots were also performed to assess total muscle O-GlcNAc-modified proteins. Data are mean ± standard deviation. Statistical analysis was performed using a 2-way ANOVA with Tukey’s post-hoc analysis. *N* = 8–10 muscles/group. *p* < 0.05 # vs. Sham (Sh); * vs. low-fat diet (LFD).

**Table 1 ijms-21-04715-t001:** Immunoblotting conditions.

Antigen	Blocking	1° Antibody	1° Antibody RRID	2° Antibody	ECL Reagent
G6PD	5% BSA	1:2000 in 5% BSA, cat#8866, lot#3, Cell Signaling Technology, Danvers, MA, USA	AB_10827744	1:2000 Rabbit-HRP	Western Lightning™
GFPT1	5% milk	1:1000 in 5% BSA, cat#14132-1-AP, Proteintech, Rosemont, IL, USA	AB_2110155	1:2000 Rabbit-HRP	Super Signal™
GLUT1	5% milk	1:4000 in 5% BSA, cat#07-1401, lot#2882724, Millipore, St. Louis, MO, USA	AB_1587074	1:2000 Rabbit-HRP	Super Signal™
GLUT3	5% milk	1:5000 in 5% BSA, cat#AB1344, lot#2943583, Millipore, St. Louis, MO, USA	AB_1587078	1:2000 Rabbit-HRP	Western Lightning™
GLUT4	5% BSA	1:2000 in 5% BSA, cat#07-1404, lot#2890837, Millipore, St. Louis, MO, USA	AB_1587080	1:2000 Rabbit-HRP	Western Lightning™
GLUT6	5% BSA	1:1000 in 5% BSA, cat#TA500639, lot#A001, OriGene Technologies, Rockville, MD, USA	AB_2270444	1:5000 Mouse-HRP	Super Signal™
GLUT10	5% milk	1:5000 in 5% BSA, cat#sc-21635, lot#G0214, Santa Cruz Biotechnology, Dallas, TX, USA	AB_10989951	1:5000 Goat-HRP	Western Lightning™
HXII	5% BSA	1:1000 in 5% BSA, cat#sc-6521, lot#H2611, Santa Cruz Biotechnology, Dallas, TX, USA	AB_648073	1:5000 Goat-HRP	Western Lightning™
O-GlcNAc	5% milk	1:2000 in 5% BSA, cat#9875, lot#4, Cell Signaling Technology, Danvers, MA, USA	AB_10950973	1:5000 Mouse-HRP	Western Lightning™

Antibodies and blotting conditions utilized in the immunoblotting analyses. All bovine serum albumin (BSA) and non-fat dry milk solutions were made by dissolving the reagent in a 1× Tris-buffered saline solution containing 0.1% Tween-20. Information on the primary antibodies utilized including dilution, solution diluted in, company name or developer, catalog number, lot number, associated Research Resource Identifiers (RRIDs) or references providing validation data are indicated in the table. The species, and dilution factor for the horseradish peroxidase (HRP) conjugated secondary antibodies are indicated in the table. All HRP-conjugated secondary antibodies were diluted in a 5% BSA, 1× TBS + 0.1% Tween-20 solution. The goat-HRP antibody was cat#PA1-28664, lot# RK2302292 from ThermoScientific (Waltham, MA, USA). The mouse-HRP antibody was cat#12-349, lot# 2365945 from Millipore (Burlington, MA, USA). The rabbit-HRP antibody was cat#PI31460 (lot# RB230194, SE251028, SH253595, and TG266717) from Fisher Scientific (Waltham, MA, USA). The enhanced chemiluminescence (ECL) substrate reagents utilized were either the Western Lightning™ Plus-ECL, cat# NEL105001EA from Perkin Elmer Life Sciences (Waltham, MA, USA) or the Super Signal™ West Femto Chemiluminescent Substrate, cat# PI34096, from Thermo Fisher Scientific, Inc. (Waltham, MA, USA). G6PD = glucose-6-phosphate dehydrogenase; GFPT1 = glutamine-fructose-6-phosphate transaminase 1; GLUT = glucose transporter; O-GlcNAc = O-linked β-*N*-acetylglucosamine.

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
