# Peer review of "Insulin Resistance Does Not Impair Mechanical Overload-Stimulated Glucose Uptake, but Does Alter the Metabolic Fate of Glucose in Mouse Muscle"

_ijms, 2020, doi:10.3390/ijms21134715_

Round 1

Reviewer 1 Report

Overall

Weyrauch et al. examined where glucose transported to the muscle goes to using overloaded and insulin resistant mouse models. Insulin resistance does not impair overload-stimulated increases in glucose transporter protein levels. Insulin resistance enhances overload-stimulated muscle glycogen accumulation, but not glycogen synthesis rate. Insulin resistance does not impair muscle glycolytic flux, or overload-stimulated muscle lactate secretion. Insulin resistance does not impair overload-stimulated activation of the pentose phosphate hexosamine biosynthetic pathway. This is an interesting study to find out where glucose goes after being taken up by muscle. However, there are several issues that need to be addressed, especially interpretation of the data.

Major comments

  1. The authors conclude that GLUT4 is not involved in the increase in glucose uptake caused by overload because GLUT4 in whole homogenate was not differ after the overload in result section. However, since GLUT4 translocates to the membrane, overload might increase GLUT4 protein content in plasma membrane, even if the whole GLUT4 remains unchanged. In fact, contraction-stimulated GLUT4 translocation and glucose uptake are not impaired by insulin resistance (PMID: 8048627). Therefore, I think that authors need to indicate no changes in GLUT4 protein in plasma membrane after overload by conducting the experiment or at least citing previous literatures.

  1. Authors conclude that higher glycogen content seen in the overload-stimulated muscles from the HFD mice is not due to an increase in glycogen synthesis rates in the result section. In the experiment, 5.5 mM glucose incubation was performed for 1 hour to measure how much glycogen was synthesized in both groups. However, in vivo blood glucose concentration in HFD was around 8 mM but that in LFD was around 5 mM (Fig.1B). Higher blood glucose concentration may induce to higher glucose uptake and glycogen synthesis, in vivo. Therefore, I think the author need to conduct in vivo experiment to determine the rate of glycogen synthesis. Alternatively, I propose to interpret the significant tendency (p = 0.058) as being a little more positive.

  1. I have two concerns about the data of muscle lactate secretion (Fig.5B). First, the reason for measuring lactate secretion is not clear. Lactate secretion is the lactate concentration in medium after the incubation, which was measured using the commercial kit. Lactate produced in the incubated muscle is secreted. There are several sources of lactate production such as exogenous glucose taken up to and endogenous glucose and glycogen in skeletal muscle. Therefore, it is not possible to verify the partition of glucose taken to muscle from lactate secretion using their experiment. Authors should describe clearly why they measured the lactate secretion.

Second, Authors interpret that higher lactate secretion means decreases in glucose oxidation in the Discussion. I have to say this is too speculative and incorrect interpretation. Glycolytic flux was determined using radioactive glucose incubation method which can validate only glucose degradation flux. However, lactate secretion was not. Increased lactate secretion is the result of increased muscle lactate concentration by activation of glycolysis and glycogenolysis. I think higher lactate release suggests higher glycogenolytic flux. Authors should amend the description about interpretation of higher lactate release from the over-loaded muscles in the Discussion.

  1. In relation to the above comment, can you use hypertrophied plantaris muscles for ex vivo incubation? I doubt that the hypertrophied muscles have become hypoxic during the incubation. This may be the reason for increased lactate release. Do you have the evidence that the hypertrophied muscles were not hypoxic during incubation?

Minor comments

  1. Authors concluded that overload did not increase glycolytic flux in muscles from either LFD or HFD mice in result section. However, same as the comment 2 above, the glucose concentration in vivo is much higher than that in the situation determined glycolytic flux ex vivo. Therefore, I think glycolytic flux is underestimated in the data obtained from the experiment. Measuring glycolytic enzyme protein or activity might support the data of glycolytic flux.

  1. I did not understand why not only glucose uptake but also glucose partition is important. I think authors need to describe why they required to study the partitioning of glucose pathway in overloaded skeletal muscle with insulin resistance.

  1. Does the muscle fiber composition change in 5 days overload? If it changes, may this affect the results of this study?

  1. Second paragraph in the Discussion is about the limitation of this study. I recommend to move second paragraph to before the final paragraph in the Discussion.

Author Response

Major comments

  1. The authors conclude that GLUT4 is not involved in the increase in glucose uptake caused by overload because GLUT4 in whole homogenate was not differ after the overload in result section. However, since GLUT4 translocates to the membrane, overload might increase GLUT4 protein content in plasma membrane, even if the whole GLUT4 remains unchanged. In fact, contraction-stimulated GLUT4 translocation and glucose uptake are not impaired by insulin resistance (PMID: 8048627). Therefore, I think that authors need to indicate no changes in GLUT4 protein in plasma membrane after overload by conducting the experiment or at least citing previous literatures.

We agree with the reviewer that the difference between total muscle GLUT4 vs. plasma membrane GLUT4 is an important distinction. Our group has recently published a study using muscle-specific GLUT4 knockout mice that demonstrated that GLUT4 is not necessary for overload-stimulated increases in skeletal muscle glucose uptake (McMillin, 2017). This reference is cited in the revised document in the Results section, subsection 2.3.

Reference: McMillin SL, Schmidt DL, Kahn BB and Witczak CA. Glucose transporter 4 (GLUT4) is not required for overload-induced glucose uptake or growth in mouse skeletal muscle. Diabetes, 66(6):1491-1500, 2017.

  1. Authors conclude that higher glycogen content seen in the overload-stimulated muscles from the HFD mice is not due to an increase in glycogen synthesis rates in the result section. In the experiment, 5.5 mM glucose incubation was performed for 1 hour to measure how much glycogen was synthesized in both groups. However, in vivo blood glucose concentration in HFD was around 8 mM but that in LFD was around 5 mM (Fig.1B). Higher blood glucose concentration may induce to higher glucose uptake and glycogen synthesis, in vivo. Therefore, I think the author need to conduct in vivo experiment to determine the rate of glycogen synthesis. Alternatively, I propose to interpret the significant tendency (p = 0.058) as being a little more positive.

We thank the reviewer for this opportunity to provide additional information regarding our study design. The glucose uptake and glucose metabolism measurements presented in this study were generated using an ex vivo skeletal muscle model system. This system was chosen because it provides the ability to control substrate availability (i.e. extracellular glucose concentrations) while eliminating the possible contribution of systemic factors (e.g. changes in blood flow, neuronal activity or circulating hormones such as insulin). Thus, by design this model does not fully recapitulate the multitude of factors present in the in vivo environment, but instead allowed us to directly assess the contribution of overload to the changes in muscle glucose metabolism.

The 5.5 mM glucose concentration was utilized to assess glycolytic flux and glycogen synthesis rates in muscles from both the LFD and HFD mice to normalize the extracellular glucose concentration across all treatment groups.  We agree with the reviewer that it would be very interesting to examine whether different concentrations of extracellular glucose would elicit different outcomes on muscle glucose partitioning between the sham and overload-stimulated muscles from both the LFD and HFD mice.  Unfortunately, performing these additional experiments is beyond the scope of the present study.

We agree with the reviewer that it is a disappointing finding that the ability of overload to stimulate glycogen synthesis was not quite statistically significant (P=0.058) in the muscles from the HFD mice.  When we performed a power analysis to estimate the sample size needed to detect a statistically significant different between the glycogen synthesis rates in the muscles from the sham, HFD group compared to the overload-stimulated, HFD group we calculated that 52 muscles/group would be needed with an α-value = 0.05 and power value = 0.8 using a 2-sided test.  When we altered the parameters to a 1-sided test, the number of muscles needed to achieve a statistically significant outcome decreased to 41 muscles/group. This statistical analysis provided strong rationale for us to not perform additional glycogen synthesis experiments.  Based on this finding we feel confident in concluding that with the conditions utilized in these experiments that overload was not increasing glycogen synthesis rates in the muscles from the HFD mice.

  1. I have two concerns about the data of muscle lactate secretion (Fig.5B). First, the reason for measuring lactate secretion is not clear. Lactate secretion is the lactate concentration in medium after the incubation, which was measured using the commercial kit. Lactate produced in the incubated muscle is secreted. There are several sources of lactate production such as exogenous glucose taken up to and endogenous glucose and glycogen in skeletal muscle. Therefore, it is not possible to verify the partition of glucose taken to muscle from lactate secretion using their experiment. Authors should describe clearly why they measured the lactate secretion.

            We thank the reviewer for the opportunity to provide additional information regarding our rationale to measure muscle lactate secretion in this study. Muscle lactate secretion was measured as an additional indicator of glycolytic flux, as the pyruvate produced by glycolysis can be converted to lactate by lactate dehydrogenase and then transported out of the muscle cell by the monocarboxylate transporter.  In addition, we agree with the reviewer that determining the origin of the lactate secreted from the overload-stimulated muscles is an important aspect of the larger picture of how glucose metabolism is altered in the overload-stimulated muscles. Unfortunately, since the [3H]-label from the radioactive glycolytic flux experiment is released as water during the enolase reaction, we were not able to determine how much of the lactate secreted originated from immediate glucose entry vs. glycogen stores. We have now amended the Results section 2.5 to clarify this point as follows:

Original sentence: “To further assess glucose utilization via glycolysis, buffer lactate levels were measured as an indicator of muscle lactate secretion.”

Revised sentence: “To further assess glucose utilization via glycolysis downstream of the enolase reaction, buffer lactate levels were measured as an indicator of muscle lactate secretion.”

  1. Second, Authors interpret that higher lactate secretion means decreases in glucose oxidation in the Discussion. I have to say this is too speculative and incorrect interpretation. Glycolytic flux was determined using radioactive glucose incubation method which can validate only glucose degradation flux. However, lactate secretion was not. Increased lactate secretion is the result of increased muscle lactate concentration by activation of glycolysis and glycogenolysis. I think higher lactate release suggests higher glycogenolytic flux. Authors should amend the description about interpretation of higher lactate release from the over-loaded muscles in the Discussion.

            We agree with the reviewer that determining the origin of the lactate secreted from the overload-stimulated muscles is an important aspect of the larger picture of how glucose metabolism is altered in the overload-stimulated muscles. Unfortunately, since the [3H]-label from the radioactive glycolytic flux experiment is released as water during the enolase reaction, we were not able to determine how much of the lactate secreted originated from immediate glucose entry vs. glycogen stores. To avoid the possibility that we could mislead a reader of this study with our speculation, we have removed the following original sentence from the Discussion section:

Sentence removed from the Discussion section: “If this speculation is true, this result would be very surprising and could suggest that an alternative substrate (i.e. lipid) is being oxidized to fuel the metabolic demands of muscle hypertrophic growth.”

In relation to the above comment, can you use hypertrophied plantaris muscles for ex vivo incubation? I doubt that the hypertrophied muscles have become hypoxic during the incubation. This may be the reason for increased lactate release. Do you have the evidence that the hypertrophied muscles were not hypoxic during incubation?

We appreciate the opportunity to provide additional information on our ex vivo mouse muscle incubation model system.  Prior to performing any of the muscle incubation studies presented in this paper, we performed a pilot experiment to examine the amount of plantaris muscle hypertrophy achieved in response to 5 days of overload in male C57BL6J mice fed a 60%kcal high fat diet for 12 weeks.  In this pilot study, we measured the width of the muscles and found that they did not hypertrophy greater than the previously reported oxygen diffusion limitation (>1.2 mm muscle diffusion thickness) (Barclay, 2005).  Thus, we do not believe that hypoxia was a contributing factor for the increase in lactate released from the overload-stimulated muscles duration the incubations in this study.

Reference: Barclay CJ.  Modelling diffusive O2 supply to isolated preparations of mammalian skeletal and cardiac muscle. Journal of Muscle Research and Cell Motility, 26:225-235, 2005.

Minor comments

Authors concluded that overload did not increase glycolytic flux in muscles from either LFD or HFD mice in result section. However, same as the comment 2 above, the glucose concentration in vivo is much higher than that in the situation determined glycolytic flux ex vivo. Therefore, I think glycolytic flux is underestimated in the data obtained from the experiment. Measuring glycolytic enzyme protein or activity might support the data of glycolytic flux.

We thank the reviewer for this opportunity to provide additional information regarding our study design. The 5.5 mM glucose concentration was utilized to assess glycolytic flux and glycogen synthesis rates in muscles from both the LFD and HFD mice to normalize the extracellular glucose concentration across all treatment groups.  We agree with the reviewer that it would be very interesting to examine whether different concentrations of extracellular glucose would elicit different outcomes on muscle glucose partitioning between the sham and overload-stimulated muscles from both the LFD and HFD mice.  Unfortunately, performing these additional studies is beyond the scope of the present study.

I did not understand why not only glucose uptake but also glucose partition is important. I think authors need to describe why they required to study the partitioning of glucose pathway in overloaded skeletal muscle with insulin resistance.

We apologize to the reviewer for not emphasizing the significance of the partitioning of glucose in skeletal muscle. We have amended the second paragraph of the Introduction to highlight the importance of glucose metabolism in the control of skeletal muscle glucose uptake. 

Introduction original sentences: “Glucose uptake into skeletal muscle is regulated by the presence of cell surface glucose transporters as well as its metabolic fate, including: 1) storage as glycogen; 2) flux via glycolysis; 3) the pentose phosphate pathway; and 4) the hexosamine pathway. While studies have linked excessive muscle glycogen accumulation and increased glucose flux through the hexosamine pathway with the development of insulin resistance [11-15]; others have linked increased glycolytic flux, and activation of the pentose phosphate pathway with enhanced cellular growth [16-19].”

Introduction revised sentences: “Glucose uptake into skeletal muscle is regulated by the presence of cell surface glucose transporters as well as its metabolic fate, including: 1) storage as glycogen; 2) flux via glycolysis; 3) the pentose phosphate pathway; and 4) the hexosamine pathway. Importantly, studies have linked excessive muscle glycogen accumulation and increased glucose flux through the hexosamine pathway with the development of muscle insulin resistance [11–15], highlighting the importance of glucose metabolism in regulating future increases in muscle glucose uptake. In addition, changes in glucose metabolism such as increased glycolytic flux and activation of the pentose phosphate pathway have been linked with enhanced cellular growth in a variety of cell types [16–19].”

Does the muscle fiber composition change in 5 days overload? If it changes, may this affect the results of this study?

            We appreciate the reviewer’s question regarding whether 5 days of overload altered the fiber type composition of the mouse plantaris muscles.  Previous work has demonstrated that 10-14 days of mechanical overload induces an increase in the cross-sectional area of all four fiber types found in mouse plantaris muscle, including type 1, type 2A, type 2B and type 2X fibers [Dunn, 1999; Goodman, 2012], suggesting the lack of a striking shift in muscle fiber type.  Thus, we do not believe that a shift in muscle fiber-type accounts for the changes observed in this study.

References:

Dunn SE, Burns JL, Michel RN. Calcineurin is required for skeletal muscle hypertrophy. JBC. 274(31):21908-21912, 1999.

Goodman CA, Kotecki JA, Jacobs BL, Hornberger TA.  Muscle fiber type-dependent differences in the regulation of protein synthesis. PLoS One. 7(5):e37890, 2012.

Second paragraph in the Discussion is about the limitation of this study. I recommend to move second paragraph to before the final paragraph in the Discussion. 

            We have now moved the original second paragraph in the Discussion section to the end of the Discussion section as suggested by the reviewer.

Reviewer 2 Report

This manuscript by Weyrauch, McMillin and Witczak investigates the mechanisms underlying mechanical overload-stimulated glucose uptake in insulin-resistant mouse muscle. Overall this manuscript is well-written with most figures clearly presented. This study addresses a clear knowledge gap and utilizes a range of complementary physiological and biochemical approaches to uncover several novel insights into these mechanosensitive mechanisms that are of broad interest to the field. To improve the manuscript’s overall clarity for the wide readership of IJMS and help strengthen the authors’ conclusions drawn from their results, major and minor comments to the authors are provided below that include several suggested figure and text revisions.

Major Comments:

1. Introduction, Paragraph 3: In line with the primary purpose of the study outlined in Lines 54-56, consider including more specific hypothesis/hypotheses in the last paragraph of the Introduction to better set the stage for the description of the study’s main findings in this paragraph and the series of subsequent measurements assessing differential partitioning of glucose metabolism. The authors’ hypotheses are currently listed in the Discussion section (e.g. Lines 252 and 274), and as currently written the overall lack of specific hypothesis/hypotheses in the Introduction suggests that this study is exploratory and not hypothesis driven.

2. Results, Line 67 and Materials and Methods: As the authors only used male mice in this study, at least include a statement reporting why only male mice were used in this study is warranted in the Materials and Methods section. Also consider potentially including a statement regarding whether similar findings were detected in female mice.

3. Figures and associated Results: In Figures showing condition effects (i.e. sham versus overload in muscles from LFD and HFD mice), overall the descriptions of statistical relationships between groups should be more clearly and specifically described in the Results by stating ‘significantly increased/decreased versus …’ and/or specifying the observed group and condition differences as ‘P<0.05 versus …’ in line with the respective figures.

For example, significant differences in percentage changes could be described in more detail to specify the sham group (LFD or HFD) being compared or by stating ‘versus respective sham’ in the Results and associated figure legend. In the present version of the manuscript it is not clear whether only the respective LFD/HFD sham group or both sham groups are being compared when describing results from ANOVA and post-hoc testing. Moreover, when a significant difference in percentage changes between LFD and HFD overload-stimulated groups is identified (e.g. Line 87 stating ‘increase in muscle weight was 2% greater in HFD mice’), this statistical relationship in the Results should be described in more detail by stating ‘P<0.05 versus LFD OVL’, in line with the statistical relationships that are reported in the respective figures.

4. Figure 3C: The reported molecular weight for the indicated GLUT4 band is not consistent between Figure 3C and the associated unpublished raw immunoblot images. The appropriate band for GLUT4 should be just above the 50 KDa marker if these raw immunoblot markers are shown correctly. Furthermore, the band indicated for GLUT4 appears to be over-exposed and may potentially be outside of the immunoblot imaging system’s dynamic range, which may have limited the authors’ ability to detect differences between groups. Loading 40 μg protein from mouse muscle lysates is excessive, and GLUT4 should be detectable in mouse muscle with less than half of this protein amount. Therefore, the authors should consider re-running GLUT4 immunoblots using < 20 μg to improve clarity of the GLUT4 band detected and potentially improve their ability to detect OVL- and HFD-induced changes.

5. Figure 4B: One of the main conclusions drawn by the authors is that insulin resistance ‘does impair over-load stimulated increases in muscle glycogen synthesis’ in the Discussion in Lines 283-284. However, the observed strong trend (P=0.058) and level of OVL-induced glycogen synthesis in muscle from HFD mice appears to be increased to a similar extend as muscle from LFD mice, suggesting this measurement may be just be due to insufficient statistical power. Therefore, the authors should: (a) consider measuring glycogen synthesis in additional muscle samples to increase statistical power and confirm/refute this main conclusion (if additional samples are available and the current situation permits these analyses); and/or (b) tone down this overall conclusion in the last paragraph of the Introduction and first/last paragraph of the Discussion if additional experiments are not possible at this time.

Minor Comments:

1. Abstract, Lines 26 and 30: Consider re-phrasing ‘by insulin resistance’ in both lines to a phrase such as ‘in insulin-resistant muscle’ to be more specific and improve clarity for the wide readership of IJMS.

2. Figure 1 Legend: Include the statement ‘Data are mean ± standard deviation’ in this legend to improve clarity, similarly to what is listed in the other figure legends in the manuscript.

3. Figure 1, 2 and 5, y-axis labels: In Figure panels 1E, 2B and 5A, please un-bold or change the font of the first character on the y-axis label in these figure panels so this first character is more clearly displayed.

4. Results, Line 130: The authors should consider re-phrasing ‘incubated with [5-3H]-glucose and [5-3H]-glycogen levels assessed’ to more specifically describe how radiolabeled glucose incubation permits assessment of glycogen synthesis levels, in order to improve clarity for the wide readership of IJMS who may not be familiar with this experimental approach.

5. Results, Line 173: Correct the word ‘increased’ to ‘decreased’ in line with the observed overload-induced reductions in NADP:NADPH ratio shown in Figure 6E.

6. Discussion, Line 229: Change the word ‘gender’ to the more appropriate term ‘sex’ or phrase ‘the sex of mice’.

7. Discussion, Lines 269-270: The present study and one of the two publications cited in this sentence do not assess physiologically-relevant effects of ‘exercise’ on muscle glucometabolic pathways. Therefore, revise the term ‘exercise-responsive’ to tone down this statement, and instead use a more appropriate phrase such as ‘exercise- and overload-responsive’ or ‘muscle contraction- and overload-responsive’.

8. Materials and Methods, Lines 303-304: To improve transparency and maximize reproducibility of this study by the field, the authors should provide more details regarding how they specifically identified samples that ‘did not clearly segregate into a LFD, insulin-sensitive or HFD, insulin-resistant group’ and were subsequently excluded from further experiments. For example, were specific ranges or cut-offs for fasting blood glucose, fasting serum insulin and/or HOMA-IR values used in this segregation? The authors should at least comment on their segregation strategy and/or highlight any potential bias associated with this segregation strategy in the Discussion.

9. Materials and Methods, Lines 362-364: The authors should include more experimental details regarding the UPLC-MRM/MS analysis of pentose phosphate pathway metabolites to maximize reproducibility of this study. For example, consider outlining key steps in the sample preparation of frozen muscle prior to MS analysis, and specify the MS instrument model used by Creative Proteomics in these analyses.

Author Response

Major Comments:

  1. Introduction, Paragraph 3: In line with the primary purpose of the study outlined in Lines 54-56, consider including more specific hypothesis/hypotheses in the last paragraph of the Introduction to better set the stage for the description of the study’s main findings in this paragraph and the series of subsequent measurements assessing differential partitioning of glucose metabolism. The authors’ hypotheses are currently listed in the Discussion section (e.g. Lines 252 and 274), and as currently written the overall lack of specific hypothesis/hypotheses in the Introduction suggests that this study is exploratory and not hypothesis driven.

We appreciate the suggestion to include the study’s hypotheses in the Introduction. We have now amended the final paragraph in the Introduction section to include the following sentence:

Added sentence to final paragraph of Introduction: “We hypothesized that: 1) glycogen content and glycogen synthesis rates would be increased by overload, and not affected by insulin resistance; 2) glycolytic flux and pentose phosphate pathway activity would be increased by overload and not affected by insulin resistance; and 3) hexosamine pathway activity would be increased in insulin-resistant muscle, and decreased by overload.”

  1. Results, Line 67 and Materials and Methods: As the authors only used male mice in this study, at least include a statement reporting why only male mice were used in this study is warranted in the Materials and Methods section. Also consider potentially including a statement regarding whether similar findings were detected in female mice.

We appreciate the opportunity to provide additional information regarding our study design.  Previous published work has demonstrated that male mice are more susceptible to the development of high fat diet-induced obesity, glucose intolerance and insulin resistance compared to female mice [Garg, 2011; Hong, 2009; Pettersson, 2012; Salinero, 2018].  Thus, to assess the ability of overload to stimulate muscle glucose uptake and glucose metabolism in an insulin resistant model, we only used male mice in this study.  We have now added the following statement to the Materials and Methods section 4.1.:

Added sentence: “Only male mice were utilized in this study because previous work has indicated that male mice exhibit greater obesity, glucose intolerance and insulin resistance in response to high fat feeding compared to female mice [Garg, 2011; Hong, 2009; Pettersson, 2012; Salinero, 2018].”

We have previously shown that the ability of overload to stimulate skeletal muscle glucose uptake is equally effective in both male and female insulin-sensitive mice [McMillin, 2017].     

References:

Garg N, Thakur S, Mcmahan CA, Adamo ML. High fat diet induced insulin resistance and glucose intolerance is gender-specific in IGF-1R heterozygous mice. Biochem Biophys Res Commun. 413(3):476-480, 2011.

Hong J, Stubbins RE, Smith RR, Harvey AE, Nunez NP. Differential susceptibility to obesity between male, female and ovariectomized female mice. Nutr J. 8:11, 2009.

McMillin SL, Schmidt DL, Kahn BB and Witczak CA. Glucose transporter 4 (GLUT4) is not required for overload-induced glucose uptake or growth in mouse skeletal muscle. Diabetes. 66(6):1491-1500, 2017

Pettersson US, Walden TB, Carlsson P-O, Jansson L, Phillipson M. Female Mice are Protected against High-Fat Diet Induced Metabolic Syndrome and Increase the Regulatory T Cell Population in Adipose Tissue. PLoS One. 7(9):e46057, 2012.

Salinero A.E., Anderson B.M., Zuloaga K.L. Sex differences in the metabolic effects of diet-induced obesity vary by age of onset. Int. J. Obes. (Lond.). 42:1088–1091. 2018.

  1. Figures and associated Results: In Figures showing condition effects (i.e. sham versus overload in muscles from LFD and HFD mice), overall the descriptions of statistical relationships between groups should be more clearly and specifically described in the Results by stating ‘significantly increased/decreased versus …’ and/or specifying the observed group and condition differences as ‘P<0.05 versus …’ in line with the respective figures.

For example, significant differences in percentage changes could be described in more detail to specify the sham group (LFD or HFD) being compared or by stating ‘versus respective sham’ in the Results and associated figure legend. In the present version of the manuscript it is not clear whether only the respective LFD/HFD sham group or both sham groups are being compared when describing results from ANOVA and post-hoc testing. Moreover, when a significant difference in percentage changes between LFD and HFD overload-stimulated groups is identified (e.g. Line 87 stating ‘increase in muscle weight was 2% greater in HFD mice’), this statistical relationship in the Results should be described in more detail by stating ‘P<0.05 versus LFD OVL’, in line with the statistical relationships that are reported in the respective figures.

We apologize to the reviewer for the lack of clarity in the description of some of our findings in the Results section.  We have now amended the following sentences:

Results section 2.2 original: “Overload increased plantaris muscle weight ~40% in LFD mice, and this increase in muscle weight was ~2% greater in HFD mice (Fig. 2A).”

Results section 2.2 amended: Overload increased plantaris muscle weight ~40% in LFD mice, and this overload-stimulated increase in muscle weight was ~2% greater in HFD mice (Fig. 2A).

Results section 2.3 original: In contrast, overload increased GLUT3 levels ~65% in muscles from LFD mice, and this effect was enhanced up to ~100% by the HFD (Fig. 3B).

Results section 2.3 amended: Overload increased GLUT3 levels ~65% in muscles from LFD mice, and increased GLUT3 levels more in muscles from the HFD mice (~100%; Fig. 3B).

Results section 2.4 original: Overload increased glycogen content ~25% in muscles from LFD mice and ~40% in muscles from HFD mice (Fig. 4A), demonstrating that lower glycogen content is not part of the mechanism underlying overload-stimulated glucose uptake.

Results section 2.4 amended: Overload increased glycogen content ~25% in muscles from LFD mice and this increase was greater (~40%) in muscles from HFD mice (Fig. 4A), demonstrating that lower glycogen content is not part of the mechanism underlying overload-stimulated glucose uptake.

Results section 2.6 original: As shown in Fig. 6A, overload increased G6PD levels ~140% in muscles from LFD mice and ~80% in muscles from HFD mice.

Results section 2.6 amended: As shown in Fig. 6A, overload increased G6PD levels ~140% in muscles from LFD mice and this effect was attenuated in muscles from HFD mice (only a ~80% increase).

Results section 2.7 original: As shown in Fig. 7B, overload increased protein O-GlcNAcylation ~45% in muscles from LFD mice, and this effect was increased up to ~55% by the HFD.

Results section 2.7 amended: As shown in Fig. 7B, overload increased protein O-GlcNAcylation ~45% in muscles from LFD mice, and this effect was greater in muscles from the HFD mice (increased ~55%). 

Throughout the manuscript we only used the word “increased” or “decreased” when the findings were statistically significant (P<0.05) as described in the Materials and Methods section 4.10s ‘Statistical Analyses’.  We did not want to repeat the word “significantly” or provide P-values in the text to simplify the reading of the study. When a finding was not statistically significant, we used language to highlight this important distinction. For example, in the Results section 2.4, we wrote the following: “In response to overload, glycogen synthesis increased ~25% in muscles from LFD mice, but only trended towards a significant increase (~12%; P=0.058) in muscles from HFD mice (Fig. 4B).”

  1. Figure 3C: The reported molecular weight for the indicated GLUT4 band is not consistent between Figure 3C and the associated unpublished raw immunoblot images. The appropriate band for GLUT4 should be just above the 50 KDa marker if these raw immunoblot markers are shown correctly. Furthermore, the band indicated for GLUT4 appears to be over-exposed and may potentially be outside of the immunoblot imaging system’s dynamic range, which may have limited the authors’ ability to detect differences between groups. Loading 40 μg protein from mouse muscle lysates is excessive, and GLUT4 should be detectable in mouse muscle with less than half of this protein amount. Therefore, the authors should consider re-running GLUT4 immunoblots using < 20 μg to improve clarity of the GLUT4 band detected and potentially improve their ability to detect OVL- and HFD-induced changes.

We apologize for our error in the reported molecular weight on the unpublished raw immunoblot GLUT4 images.  The GLUT4 protein is located between the 50 kDa and 37 kDa molecular weight standards using our immunoblotting procedures.  We have now made this correction on the raw immunoblot GLUT4 images. We also appreciate the opportunity to provide additional information on our immunoblot analyses.  The BioRad Chemidoc XRS+ imager and ImageLab software enabled us to capture numerous digital images with different exposure times for each membrane analyzed.  If any portion of the GLUT4 protein band became over-exposed/saturated, the ImageLab software would change the color of that pixel from gray scale to red allowing us to easily detect the saturated pixel. Only images in which none of the pixels comprising the GLUT4 band were over-exposed/saturated were utilized in the quantification of GLUT4.  Thus, we feel very confident that the lack of a significant change in GLUT4 protein levels in response to overload is the correct finding for this study.  It is also consistent with our previous work demonstrating no significant increases in total muscle GLUT4 protein levels in response to 5 days of overload (McMillin, 2017).

Reference: McMillin SL, Schmidt DL, Kahn BB and Witczak CA. Glucose transporter 4 (GLUT4) is not required for overload-induced glucose uptake or growth in mouse skeletal muscle. Diabetes, 66(6):1491-1500, 2017.

  1. Figure 4B: One of the main conclusions drawn by the authors is that insulin resistance ‘does impair over-load stimulated increases in muscle glycogen synthesis’ in the Discussion in Lines 283-284. However, the observed strong trend (P=0.058) and level of OVL-induced glycogen synthesis in muscle from HFD mice appears to be increased to a similar extend as muscle from LFD mice, suggesting this measurement may be just be due to insufficient statistical power. Therefore, the authors should: (a) consider measuring glycogen synthesis in additional muscle samples to increase statistical power and confirm/refute this main conclusion (if additional samples are available and the current situation permits these analyses); and/or (b) tone down this overall conclusion in the last paragraph of the Introduction and first/last paragraph of the Discussion if additional experiments are not possible at this time.

We agree with the reviewer that it is a disappointing finding that the ability of overload to stimulate glycogen synthesis was not quite statistically significant (P=0.058) in the muscles from the HFD mice.  When we performed a power analysis to estimate the sample size needed to detect a statistically significant different between the glycogen synthesis rates in the muscles from the sham, HFD group compared to the overload-stimulated, HFD group we calculated that 52 muscles would be needed with an α-value = 0.05 and power value = 0.8 using a 2-sided test.  When we altered the parameters to a 1-sided test, the number of muscles needed to achieve a statistically significant decreased to 41 muscles/group. This statistical analysis provided strong rationale for us to not perform additional glycogen synthesis experiments.  Based on this finding we feel confident in concluding that with the conditions utilized in these experiments that overload was not increasing glycogen synthesis rates in the muscles from the HFD mice.

Minor Comments:

  1. Abstract, Lines 26 and 30: Consider re-phrasing ‘by insulin resistance’ in both lines to a phrase such as ‘in insulin-resistant muscle’ to be more specific and improve clarity for the wide readership of IJMS.

We have now amended lines 26 and 30 in the abstract to include the use of the phrase “in the insulin-resistant muscle”.

  1. Figure 1 Legend: Include the statement ‘Data are mean ± standard deviation’ in this legend to improve clarity, similarly to what is listed in the other figure legends in the manuscript.

Thank you for pointing out this omission. We have now added into figure legend 1 that the data are presented as the mean + standard deviation.  

  1. Figure 1, 2 and 5, y-axis labels: In Figure panels 1E, 2B and 5A, please un-bold or change the font of the first character on the y-axis label in these figure panels so this first character is more clearly displayed.

We apologize that the “µ” symbol found in the y-axis labels for figures 1E, 2B and 5A did not display clearly on your version of the article to review.  Unfortunately, it appears clear in our current version and so at this point there is nothing that we can do.  We will take extra care in looking for this problem in the final version of the document.

  1. Results, Line 130: The authors should consider re-phrasing ‘incubated with [5-3H]-glucose and [5-3H]-glycogen levels assessed’ to more specifically describe how radiolabeled glucose incubation permits assessment of glycogen synthesis levels, in order to improve clarity for the wide readership of IJMS who may not be familiar with this experimental approach.

            We apologize for the lack of clarity in the description of the glycogen synthesis methods.  We also identified an error in this sentence in the statement that the radiolabeled glucose incubation occurred for 2 hrs when it only occurred for 1 hr. We have now amended the text to improve clarity of this sentence for future readers of this manuscript as follows:

Results section 2.4 original sentence: “To determine whether the higher glycogen content observed in the overload-stimulated muscles was due to increased glycogen synthesis, muscles were incubated in [5-3H]-glucose and [5-3H]-glycogen levels assessed 2 hours later.”

Results section 2.4 amended sentence: “To determine whether the higher glycogen content observed in the overload-stimulated muscles was due to increased glycogen synthesis, muscles were incubated in [5-3H]-glucose for 1 hour, and the incorporation of the radiolabel into glycogen (i.e. [5-3H]-glycogen) assessed.”

  1. Results, Line 173: Correct the word ‘increased’ to ‘decreased’ in line with the observed overload-induced reductions in NADP:NADPH ratio shown in Figure 6E.

            We apologize for this error. We have now corrected the sentence to indicate the decrease in the ratio of NADP:NADPH.

  1. Discussion, Line 229: Change the word ‘gender’ to the more appropriate term ‘sex’ or phrase ‘the sex of mice’.

We have now changed the manuscript and utilized the term “the sex of mice” instead of “gender”.

  1. Discussion, Lines 269-270: The present study and one of the two publications cited in this sentence do not assess physiologically-relevant effects of ‘exercise’ on muscle glucometabolic pathways. Therefore, revise the term ‘exercise-responsive’ to tone down this statement, and instead use a more appropriate phrase such as ‘exercise- and overload-responsive’ or ‘muscle contraction- and overload-responsive’.

We thank the reviewer for their comment and have now changed the manuscript and to include the phrase ‘exercise- and overload-responsive’ as follows:

Discussion section original sentence: “Thus, collectively these findings support the idea that the pentose phosphate pathway is an exercise-responsive glucometabolic pathway in muscle.”

Discussion section revised sentence: “Thus, collectively these findings support the idea that the pentose phosphate pathway is an exercise- and overload-responsive glucometabolic pathway in muscle.”

  1. Materials and Methods, Lines 303-304: To improve transparency and maximize reproducibility of this study by the field, the authors should provide more details regarding how they specifically identified samples that ‘did not clearly segregate into a LFD, insulin-sensitive or HFD, insulin-resistant group’ and were subsequently excluded from further experiments. For example, were specific ranges or cut-offs for fasting blood glucose, fasting serum insulin and/or HOMA-IR values used in this segregation? The authors should at least comment on their segregation strategy and/or highlight any potential bias associated with this segregation strategy in the Discussion.

            We appreciate the opportunity to provide additional information regarding our study design, and specifically the approach that we utilized to identify LFD and HFD mice that had responded to the diet treatments prior to inclusion in our study.  Unfortunately, there are currently no specific criteria/reference ranges that exist to define type 2 diabetes or insulin resistance in mice. Thus, we had to establish our own inclusion/exclusion criteria by looking at measures that were not terminal to the mice. 

After 12 weeks on the diets, mice were fasted overnight and blood collected to assess glucose and insulin levels.  Using these blood values, we calculated HOMA-IR.  (These values are now provided in the tables below and are segregated into cohorts based on the final measurements made the plantaris muscles.  For example, one cohort was only utilized to assess insulin-stimulated muscle glucose uptake.)  Using these values, we included the LFD mice that exhibited the lowest of these measurements, and the HFD mice with the highest of these measurements within each cohort of mice.  Importantly, the cohort utilized to directly assess skeletal muscle insulin resistance (i.e. insulin-stimulated muscle glucose uptake) exhibited the lowest change in fasted blood glucose levels, and one of the lowest changes in fasted serum insulin and HOMA-IR values of all of the groups.  Based on this information, we feel confident that we can state that this study examined the effects of overload on insulin-resistant skeletal muscle.

  1. Materials and Methods, Lines 362-364: The authors should include more experimental details regarding the UPLC-MRM/MS analysis of pentose phosphate pathway metabolites to maximize reproducibility of this study. For example, consider outlining key steps in the sample preparation of frozen muscle prior to MS analysis, and specify the MS instrument model used by Creative Proteomics in these analyses.

We have now included additional methodological details regarding the UPLC-MRM/MS analysis of the pentose phosphate pathway metabolites in the Material and Methods section 4.8. as indicated below:

Information added to Materials and Methods section 4.8.: “Whole frozen plantaris muscles were sent to Creative Proteomics (Shirley NY) for the quantification of NADP, NADPH, and 6-phosphogluconate levels. Muscles were weighed and then homogenized in cold 80% aqueous MeOH using a MM 400 mill mixer 2x1 min. Samples were centrifuged at 21,130 xg at 5°C, and 400 µl of clear supernatant removed and mixed with 100 µl of a 13C10-GTP (internal standard) solution and 250 µl chloroform. The mixture was vortexed for 1 min and then centrifuged at 21,130 xg at 5°C for 10 min. The upper aqueous phase was removed and dried under a nitrogen gas flow. The residue was reconstituted in 200 µl of 50% acetonitrile. A 10 µl aliquot of each sample was injected into a HILIC (hydrophilic interaction liquid chromatography) column (2.1 x 150 mm, 1.7 µm) for gradient elution with ammonium buffer (mobile phase A) and acetonitrile (mobile phase B). The efficient gradient was from 75% B to 15% B in 12 min, with a flow rate of 300 µl/min and a column temperature of 30°C. Metabolites were detected by UPLC-MRM/MS with (-) ion detection using a Dionex 3400 UHPLC system coupled to a 4000 QTRAP mass spectrometer.” 

Reviewer 3 Report

Weyrauch et al. have provided in this manuscript novel information regarding glucose metabolism in hypertrophying skeletal muscle under standard and insulin resistant conditions. Their primary findings are that glucose uptake is not altered by insulin resistance under overload conditions, but the cell utilization of the glucose is affected. The paper is well-written and clear, and the experiments appear to be well-executed. Nonetheless, I think the following points should be addressed to strengthen the manuscript.

  1. The authors used a 5 day overload procedure. Was there a reason for choosing this particular timepoint? At 5 days the plantaris is likely still very inflamed (probably contributing somewhat to the increased mass with overload). Furthermore, the increased inflammation likely decreases oxidative metabolism and increases glycolytic flux (e.g. see Frisard et al., DOI: 10.1016/j.metabol.2014.11.007). As the authors note in line 256, it would appear that overload is decreasing oxidation of glucose (and fat?), but this might be secondary to the inflammation. This makes me wonder if the results of the current study would be different in muscle at day 7 or 14 of overload when the inflammatory burden has resolved for the most part. Furthermore, 12 weeks of HFD  may or may not alter this inflammatory response such that additional differences might be observed with the overload-stimulated inflammation had died down. Therefore, the addition of a couple of markers of inflammation in the muscles would be helpful in better understanding the metabolic observations in the context of mechanical overload-hypertrophy and inflammatory response, and a discussion of this might be added to the manuscript.
  2. The authors report that muscle hypertrophy (as gauged by overall muscle mass) was not affected by HFD induced insulin resistance. This is in contrast to previous findings (e.g. Sitnick et al., doi: 10.1113/jphysiol.2009.180174) where muscle mass hypertrophy was attenuated by HFD, but only at later timepoints. Protein synthesis in that study (at least translation) was substantially decreased at 3 and 7 days. In the current study, since muscle glycogen content was elevated in the HFD-fed muscles (along with the carbohydrate-associated water), this might indicate that protein accretion was, in fact, reduced in the HFD muscles. Was protein concentration affected by HFD? A brief discussion of this point might be made to reconcile these results with previous work.
  3. At least in my review copy, some of the figures (for example 5A) were a little chaotic with the individual data points and error bars all superimposed over the bar graph. While I Iike seeing the data points, it makes it hard to see the actual mean. Can the design of these graphs be improved? I would leave that to the author’s discretion, however as it is not a major issue.

Author Response

Reviewer 3:

The authors used a 5 day overload procedure. Was there a reason for choosing this particular timepoint? At 5 days the plantaris is likely still very inflamed (probably contributing somewhat to the increased mass with overload). Furthermore, the increased inflammation likely decreases oxidative metabolism and increases glycolytic flux (e.g. see Frisard et al., DOI: 10.1016/j.metabol.2014.11.007). As the authors note in line 256, it would appear that overload is decreasing oxidation of glucose (and fat?), but this might be secondary to the inflammation. This makes me wonder if the results of the current study would be different in muscle at day 7 or 14 of overload when the inflammatory burden has resolved for the most part. Furthermore, 12 weeks of HFD  may or may not alter this inflammatory response such that additional differences might be observed with the overload-stimulated inflammation had died down. Therefore, the addition of a couple of markers of inflammation in the muscles would be helpful in better understanding the metabolic observations in the context of mechanical overload-hypertrophy and inflammatory response, and a discussion of this might be added to the manuscript.

We thank the reviewer for his/her comment and appreciate this opportunity to provide additional information on our study design.  We performed 5 days of overload in this study to ensure that the size of the overload-stimulated muscles did not exceed the oxygen diffusion limit (>1.2 mm muscle diffusion thickness) of the whole muscle such that the innermost muscle fibers did not become hypoxic (Barclay, 2005).  We do not believe that the plantaris muscles were experiencing inflammation at 5 days post-overload as previous work has shown that in rat soleus muscle 3 days of overload did not increase the volume of ED1+ or ED2+ macrophages (Thompson, 2006), and that in rat extensor digitorum longus muscle 1 week of overload did not elicit any detectable histological evidence of inflammation (inflammatory cell infiltration or interstitial edema) (Zhou et al, 1998). 

References:

Barclay CJ.  Modelling diffusive O2 supply to isolated preparations of mammalian skeletal and cardiac muscle. Journal of Muscle Research and Cell Motility, 26:225-235, 2005.

Thompson RW, McClung JM, Baltgalvis KA, Davis JM, Carson JA. Modulation of overload-induced inflammation by aging and anabolic steroid administration. Experimental Gerontology, 41(11):1136-1148, 2006

Zhou AL, Egginton S, Brown MD, and Hudlicka O. Capillary Growth in Overloaded, Hypertrophic Adult Rat Skeletal Muscle: An Ultrastructural Study. The Anatomical Record, 252:49-63, 1998.

The authors report that muscle hypertrophy (as gauged by overall muscle mass) was not affected by HFD induced insulin resistance. This is in contrast to previous findings (e.g. Sitnick et al., doi: 10.1113/jphysiol.2009.180174) where muscle mass hypertrophy was attenuated by HFD, but only at later timepoints. Protein synthesis in that study (at least translation) was substantially decreased at 3 and 7 days. In the current study, since muscle glycogen content was elevated in the HFD-fed muscles (along with the carbohydrate-associated water), this might indicate that protein accretion was, in fact, reduced in the HFD muscles. Was protein concentration affected by HFD? A brief discussion of this point might be made to reconcile these results with previous work.

We appreciate this opportunity to provide additional information regarding our findings.  We agree with the reviewer that the Sitnick et al. paper (JPhysiol, 2009) reported no impairment in overload-stimulated plantaris muscle growth between LFD and HFD mice at 3 and 7 days post-overload. The findings from our study observing no impairment in overload-stimulated plantaris muscle mass 5 days post-overload (Figure 2A) are consistent with these findings. To assess whether there were any differences in muscle protein content in this study, we examined the protein concentration of the muscles utilized in our immunoblot analyses. Calculating the amount of total protein isolated per mg muscle weight revealed no differences between amongst any of the treatment groups.  The protein concentrations were as follows (data presented as mean + standard deviation): LFD, sham: 93.94 + 8.51 µg protein/mg muscle; LFD, overload: 90.40 + 9.06 µg protein/mg muscle; HFD, sham: 96.86 + 7.39 µg protein/mg muscle; HFD overload: 90.24 + 6.02 µg protein/mg muscle.

At least in my review copy, some of the figures (for example 5A) were a little chaotic with the individual data points and error bars all superimposed over the bar graph. While I Iike seeing the data points, it makes it hard to see the actual mean. Can the design of these graphs be improved? I would leave that to the author’s discretion, however as it is not a major issue.

            We apologize that some of the figures were not clear with the individual data points and error bars.  We feel that including the individual data points enables the reader to see the full range of data variability instead of just providing the mean and standard deviation.  We appreciate the reviewer’s openness in letting us make this final choice in the presentation of the data.

Round 2

Reviewer 2 Report

The authors have sufficiently addressed my major and minor comments, and their revisions to the text and figures have helped strengthened the overall manuscript. I commend the authors for their attention to detail in the revisions and responses to reviewers.

This study represents a novel, high quality and scientifically sound contribution that is of interest to the field and wide readership of IJMS.